# The Prognostic Value of the Serum Level of C-Reactive Protein for Survival of Children with Ewing’s Sarcoma

**DOI:** 10.3390/cancers15051573

**Published:** 2023-03-03

**Authors:** Costantino Errani, Matteo Traversari, Monica Cosentino, Marco Manfrini, Stefano Basoli, Shinji Tsukamoto, Andreas F. Mavrogenis, Barbara Bordini, Davide Maria Donati

**Affiliations:** 1Clinica Ortopedica e Traumatologica III a Prevalente Indirizzo Oncologico, IRCCS Istituto Ortopedico Rizzoli, 40136 Bologna, Italy; 2Laboratorio di Tecnologia Medica, IRCCS Istituto Ortopedico Rizzoli, 40136 Bologna, Italy; 3Department of Orthopaedic Surgery, Nara Medical University, 840, Shijo-cho, Kashihara-city 634-8521, Japan; 4First Department of Orthopaedics, School of Medicine, National and Kapodistrian University of Athens, 41 Ventouri Street, Holargos, 15562 Athens, Greece

**Keywords:** inflammatory biomarkers, C-reactive protein, Ewing’s sarcoma, long bones, appendicular skeleton, survival, prognosis, local recurrence, children

## Abstract

**Simple Summary:**

Ewing’s sarcoma is a highly aggressive malignant small round-cell tumor that mainly affects children and adolescents. A raised level of C-reactive protein (CRP) seems to be an indicator of poor prognosis in several cancers. The purpose of the present study was to evaluate the prognostic role of CRP in children with Ewing’s sarcoma in relation to previously cited variables, including age, gender, tumor volume, tumor site, chemotherapy-induced necrosis, and other inflammatory biomarkers. The identification of reliable prognostic factors could help to identify high-risk patients, which may require a different treatment and follow up.

**Abstract:**

The purpose of the present study was to evaluate the prognostic role of C-reactive protein (CRP) in children with Ewing’s sarcoma. We conducted a retrospective study on 151 children undergoing multimodal treatment for Ewing’s sarcoma in the appendicular skeleton from December 1997 to June 2020. Univariate Kaplan-Meier analyses of laboratory biomarkers and clinical parameters showed that CRP and metastatic disease at presentation were poor prognostic factors associated with overall survival and disease recurrence at 5 years (*p* < 0.05). A multivariate Cox regression model showed that pathological CRP (≥1.0 mg/dL) [HR of 3.67; 95% CI, 1.46 to 10.42] and metastatic disease [HR of 4.27; 95% CI, 1.58 to 11.47] were associated with a higher risk of death at 5 years (*p* < 0.05). In addition, pathological CRP (≥1.0 mg/dL) [HR of 2.66; 95% CI, 1.23 to 6.01] and metastatic disease [HR of 2.56; 95% CI, 1.13 to 5.55] were associated with a higher risk of disease recurrence at 5 years (*p* < 0.05). Our findings demonstrated that CRP was associated with the prognosis of children with Ewing’s sarcoma. We recommend pre-treatment measurement of the CRP in order to recognize children with Ewing’s sarcoma who are at greater risk of death or local recurrence.

## 1. Introduction

Ewing’s sarcoma is a highly aggressive malignant small round-cell tumor that mainly affects children, adolescents and young adults. It predominantly occurs in the bones and occasionally in the soft tissues [1,2,3,4,5]. In the pre-chemotherapeutic era, the overall survival at 10 years was approximately 10% [6]. With the introduction of multimodal treatment, combining surgery and/or radiotherapy with systemic chemotherapy, the prognosis improved to 65–75% at 10 years for patients with localized disease, and 30% for those with metastatic disease [2]. Although there are no doubts that the presence of metastatic disease at presentation is the main negative prognostic factor, for patients with localized disease, the prognostic significance of several prognostic factors is still uncertain [7]. A raised level of C-reactive protein (CRP) has been found to be an indicator of poor prognosis in several cancers, including esophageal carcinoma, carcinoma of the stomach, colorectal cancer, hepatocellular carcinoma, renal cancer, and pancreatic tumor [8,9,10,11]. The prognostic value of CRP has also been evaluated in patients affected by bone and soft tissue sarcomas, without differentiating the various histotypes that have distinct treatments and prognoses [12,13,14,15]. The only studies that analyzed CRP in patients with Ewing’s sarcoma had important limitations [13,16,17,18,19]. Nakamura et al. analyzed patients without rule out the presence of inflammatory conditions [13]. Li et al. analyzed patients with Ewing’s sarcoma who underwent a heterogeneous treatment with many patients not receiving chemotherapy and the variables analyzed did not include chemotherapy-induced tumor necrosis and tumor volume [16]. Xu et al. only analyzed patients with vertebral or sacral Ewing’s sarcoma, which are usually associated with worse prognosis [17]. Recently, two other studies analyzed the possible association between pathological CRP and prognosis in patients with Ewing’s sarcoma, but the authors did not analyze tumor size, other inflammatory biomarkers, and did not consider the subgroup analysis of localized patients and metastatic patients [18,19].

The purpose of the present study was to evaluate the prognostic role of CRP in children with Ewing’s sarcoma in relation to previously cited variables, including age, gender, tumor volume, tumor site, and chemotherapy-induced necrosis. The identification of reliable prognostic factors could help to identify high-risk patients, which may require a different treatment and follow up.

## 2. Materials and Methods

An independent ethics committee approved this study, which was registered with ClinicalTrials.gov (NCT05100368).

We conducted a retrospective study of 362 patients undergoing multimodal treatment for Ewing’s sarcoma in the appendicular skeleton from December 1997 to June 2020. We investigated potential clinical and laboratory prognostic factors. The prognostic clinical and laboratory factors considered for the analysis were chosen based on previous studies on bone and soft tissue sarcoma survival factors [13,16,20,21,22].

The inclusion criteria were patients younger than 18 years old with a pathologically confirmed diagnosis of Ewing’s sarcoma of the long bones that were treated at our institution. Exclusion criteria were patients who underwent previous treatments including chemotherapy, radiotherapy and surgery (118); patients who had clinical evidence of infection or inflammatory disease before the biopsy (2); patients who underwent any major surgical treatment in the previous 30 days from the diagnosis; patients with incomplete clinical or laboratory data (83); and patients who were alive with a follow-up shorter than 24 months (8).

One hundred and fifty-one patients were eligible for the study (Figure 1). There included 40 females (26.5%) and 111 males (73.5%), with a mean age of 13 years (ranging from three to 18 years). The mean follow-up was 75 months (ranging from 24 to 224 months). The most common tumor site was the pelvis (25.8%), followed by the femur (23.8%) and the tibia (21.2%). A chest Computed Tomography and PET or bone scan were performed for disease staging. One hundred and eight patients (71.5%) had localized disease, and 43 patients (28.5%) had metastatic disease at presentation. The most common metastatic site was the lung (55.8%), followed by the bones (18.6%).

A multidisciplinary team (according to European guidelines) made decisions about treatment. Radiotherapy was considered if patients were not candidates for surgery, if there was a positive surgical margin after excision, or if the tumor had a poor response to chemotherapy (<90% tumor necrosis). Each patient underwent chemotherapy and surgical treatment in our institute but received radiotherapy at their local oncology hospitals. One hundred and fourteen patients (75.5%) underwent surgical treatment as local treatment, 34 patients (22.5%) received definitive radiotherapy as local treatment, and three patients (2%) only received chemotherapy. Neoadjuvant chemotherapy according to protocols used at the time of treatment was provided to all patients (Table 1).

The histopathological diagnosis was made in our institute. The surgical margins were wide in 106 patients (93%), marginal in four patients (3.5%), and intralesional in four patients (3.5%). The response to chemotherapy was good (tumor necrosis ≥ 90%) in 62 patients (54.4%) and poor (tumor necrosis < 90%) in 52 patients (45.6%).

We analyzed the following prognostic factors: patient age, gender, tumor site, tumor size (≥200 mL or <200 mL), metastasis at presentation, treatment methods (neoadjuvant or adjuvant chemotherapy, radiotherapy, surgery), and response to chemotherapy. We also analyzed several inflammatory biomarkers suggested in previous studies: CRP, neutrophil count, lymphocyte count, monocyte count, platelet count, serum hemoglobin, alkaline phosphatase, lactate dehydrogenase (LDH), neutrophil-lymphocyte ratio (NLR), and platelets-lymphocyte ratio (PLR) [13,16,23]. Tumor volume was calculated as an ellipsoid mass according to previous studies [2,22]. Inflammatory biomarkers were measured before biopsy (Table 2). NLR was calculated by dividing the neutrophil count by the lymphocyte count [16]; PLR was calculated by dividing the platelet count by the lymphocyte count [16].

The laboratory biomarkers were measured as part of a routine biochemical examination before any treatment, including chemotherapy, radiotherapy and surgery. We defined the cut off value for each biomarker (normal versus high or low) according to previous studies on the prognosis of bone and soft tissue sarcomas [13,16,23]. Pathological CRP was defined as values > 1 mg/100 mL [9]. A high level of NLR was defined as >2.38 [16]. The AU Beckman Coulter 680 analyzer was used for the CRP analysis and a Dasit system XN-1000 was used for the remaining laboratory biomarkers [24].

Surveillance for local and systemic recurrence consisted of clinical and radiologic assessments (radiograph of the surgical site and chest Computed Tomography) every 3 months for the first 2 years, then every 6 months up to 5 years, and annually thereafter.

### Statistical Analysis

Statistical analyses were performed with the use of JMP^®^, Version 12.0.1. SAS Institute Inc., Cary, NC, USA, 1989–2007 and R version 3.4.2. (Comprehensive R Archive Network).

Overall survival time was set as the interval between the date of diagnosis and the date of the latest follow-up or death. Recurrence-free survival time was set as the interval between the date of diagnosis and the date of the latest follow-up or local versus distant recurrence.

CRP has been used to investigate a possible association with 5-year overall survival and 5-year recurrence-free survival, separately in the group of patients with metastatic disease at presentation and in the group with localized disease. Survival curves were estimated with the Kaplan-Meier method. The log-rank test was used to compare survival between groups.

A further analysis was conducted testing all the recorded laboratory biomarkers and clinical factors in the group of patients treated with surgery as the local treatment, including both patients with metastatic and localized disease, in order to investigate a possible association with 5-year overall survival and 5-year recurrence free survival. Survival curves were estimated with the Kaplan-Meier method. The log-rank test was used to compare survivorship between groups. A multivariate Cox regression model was estimated with the parameters that were statistically significant at the univariate analysis. Hazard ratios and their corresponding 95% CI were estimated, and *p* values < 0.05 were considered significant.

## 3. Results

The 5-year overall survival was 68.5% (95% CI 60.2–75.8): 102 patients (67.5%) were alive, while 49 patients (32.5%) died. Eighty-five patients (56.3%) were alive with no evidence of disease, 13 patients (8.6%) were alive with no evidence of disease after a recurrence, and four patients (2,6%) were alive with disease.

The 5-year recurrence-free survival was 60.1% (95 CI 51.6–67.9): 60 patients (39.7%) developed a recurrence of disease (nine patients had a local recurrence, 42 patients had distant metastasis and nine patients had both local recurrence and distant metastasis).

A univariate Kaplan-Meier analysis showed that CRP was a negative prognostic factor associated with overall survival at 5 years both in patients with metastatic and localized disease, *p* = 0.010 and *p* = 0.004, respectively (Table 3). In the group of patients with localized disease, patients with pathological CRP had a poorer 5-year overall survival than patients with a normal CRP (*p* = 0.004). The overall survival at five years was 65.6% (95% CI 47.9 to 79.8) in patients with pathological CRP and 92.7% (95% CI 79.6 to 97.6) in those with normal CRP (Figure 2). In the group of patients with metastatic disease at presentation, patients with pathological CRP had a poorer 5-year overall survival than patients with normal CRP (*p* = 0.010). The overall survival at five years was 12.5% (95% CI 4.1 to 32.4) in patients with pathological CRP, and 50% (95% CI 24.4 to 75.6) in those with normal CRP (Figure 3).

CRP was a negative prognostic factor associated with recurrence-free survival at 5 years in the group of patients with localized disease (*p* = 0.024). The estimated 5-year recurrence-free survival was 54.8% (95% CI 37.4 to 71.1) in patients with pathological CRP, and 79.5% (95% CI 65.1 to 89.0) in those with normal CRP (Figure 4).

A further evaluation was performed in the group of patients treated with surgery as a major local treatment, including both patients with metastatic and localized disease (Table 4). Univariate Kaplan-Meier analyses of laboratory biomarkers and clinical parameters showed that CRP, metastatic disease at presentation, and radiotherapy were poor prognostic factors associated with overall survival at 5 years (*p* < 0.05).

A multivariate Cox regression model of these three negative prognostic factors at univariate analysis was performed (Table 5): pathological CRP (≥1.0 mg/dL) [HR of 3.67; 95% CI, 1.46 to 10.42] and metastatic disease [HR of 4.27; 95% CI, 1.58 to 11.47] were associated with a higher risk of death at 5 years (*p* < 0.05).

In addition, pathological CRP, metastatic disease, radiotherapy and NLR were associated with recurrence-free survival at 5 years (Table 6). Pathological CRP and metastatic disease remained significant in the multivariate Cox analysis for both overall survival and recurrence-free survival.

A multivariate Cox regression model of negative prognostic factors was performed (Table 7): pathological CRP (≥1.0 mg/dL) [HR of 2.66; 95% CI, 1.23 to 6.01] and metastatic disease [HR of 2.56; 95% CI, 1.13 to 5.55] were associated with a higher risk of disease recurrence at 5 years (*p* < 0.05).

## 4. Discussion

Elevated levels of inflammatory biomarkers and poor prognosis have been reported not only in patients with cancers, but also in patients with bone and soft tissue sarcomas [13,23]. Although metastatic disease at presentation is a well-known negative prognostic factor in patients with Ewing’s sarcoma, the prognostic significance of inflammatory biomarkers is still uncertain [7,25]. Our findings showed that CRP was a negative prognostic factor in children with Ewing’s sarcoma. We reported that pathological CRP and metastatic disease were both associated with overall survival and recurrence-free survival. These results may suggest that pathological CRP seems to be related to aggressive tumor behavior, with a higher risk of local recurrence and metastasis.

Our study has a few limitations. First, its retrospective nature is a major limitation. Second, although this study was performed in a single institution, patients were probably treated differently over the years. Third, we had a relatively small series of patients because of the rarity of Ewing’s sarcoma; the outcome of the study could change with a larger patient population.

Nakamura et al. retrospectively analyzed 318 patients with bone sarcomas between 2003 and 2010: pathological CRP was a negative prognostic factor of survival and local recurrence. They also analyzed the association between CRP, survival and local control for patients with Ewing’s sarcoma, showing that pathological CRP was a negative prognostic factor for survival [13]. In contrast, pathological CRP was not a negative prognostic factor for local control [13]. One of the limitations of this study is that the authors did not analyze the presence of systemic disease that may be associated with higher levels of inflammatory biomarkers [13]. Patients with sarcoma may have concurrent morbidity that causes a rise in their CRP; therefore, in our study we excluded patients who had clinical evidence of infection or inflammatory disease. Our findings showed that pathological CRP was a negative prognostic factor for both survival and disease recurrence in children with Ewing’s sarcoma.

Li et al. retrospectively analyzed 122 patients with Ewing’s sarcoma between 2009 and 2015, showing that several inflammatory prognostic biomarkers, including CRP, were correlated with the survival of patients with Ewing’s sarcoma [16]. The major limitation of this study was that the authors did not analyze the tumor size and the response to chemotherapy that are well-known prognostic factors in patients with Ewing’s sarcoma; in addition, the authors analyzed both patients with metastatic and those with localized disease as a single group [7]. In our study, we separately analyzed patients with metastatic disease and localized disease. To the best of our knowledge, this is the first study that confirms that pathological CRP is associated with poor prognosis both in patients with metastatic disease and localized disease. In addition, the tumor size and the response to chemotherapy were analyzed in patients treated surgically, confirming that pathological CRP was associated with poor survival in children with Ewing’s sarcoma.

Xu et al. retrospectively analyzed 83 patients with Ewing’s sarcoma of the spine between 2007 and 2016, reporting that the presence of metastasis and a CRP/albumin ratio < 1.5 were negative prognostic factors for overall survival. The major limitation of the study is that the authors only analyzed patients with Ewing’s sarcoma of the spine, and most patients did not receive chemotherapy prior to surgery [17]. However, we confirm these data by showing that pathologic CRP and metastatic disease are associated with overall survival in patients with Ewing’s sarcoma of not only the spine but also of the long bones.

Consalvo et al. retrospectively analyzed 40 patients with Ewing’s sarcoma, showing that CRP was different in patients with a poorer prognosis and in patients with distant metastases, whereas CRP was not different in patients with local recurrence. The major limitation of the study is that the authors did not analyze the tumor size or other inflammatory biomarkers, and the study did not consider the subgroup analysis of localized patients and metastatic patients. In addition, 13 patients did not undergo surgical treatment, and thus the relatively small number of the remaining patients may have influenced the statistical analysis [19]. Del Baldo et al. al retrospectively analyzed 89 pediatric patients with Ewing’s sarcoma, showing that LDH and CRP were associated with a poorer prognosis at univariate analysis, while only LDH remained associated with a poorer prognosis at multivariate analysis [18]. As in the previous study, the authors did not analyze tumor size, other inflammatory biomarkers, and did not consider the subgroup analysis of localized patients and metastatic patients.

In addition, Consalvo et al. and Del Baldo et al. [18,19] used 0.5 mg/dL instead of 1 mg/dL as a cutoff for physiological CRP, unlike recent literature regarding prognostic inflammatory biomarkers for musculoskeletal tumors [13,16,17,20,23,24].

Laboratory biomarkers are known to be prognostic factors for some malignancies [8,9,10]; however, very few studies have investigated the prognostic value of serum biomarkers in patients with Ewing’s sarcoma [13,16,17,18,19]. Most of these tested biomarkers are related to a systemic inflammatory process, and this is not surprising, since it is known that a systemic inflammation can be associated with cancer development and progression [20,26,27,28,29]. CRP is the only single biomarker that, according to the literature, is prognostic of both bone and soft tissue sarcomas as well as for cancer [15,20,30,31,32,33]. Avnet et al. reported that tumor acidosis could be directly associated with tumor invasion [26]. Cancer seems to induce permanent inflammation, and cancer-associated inflammation could eventually lead to tumor progression and metastasis [29].

## 5. Conclusions

Our findings showed that CRP was associated with the prognosis of children with Ewing’s sarcoma of the long bones. We recommend pre-treatment measurement of the CRP in order to recognize children with Ewing’s sarcoma who are at greater risk of death or local recurrence. Multicenter prospective studies are necessary to confirm these preliminary results in a larger patient population.

## Figures and Tables

**Figure 1 cancers-15-01573-f001:**
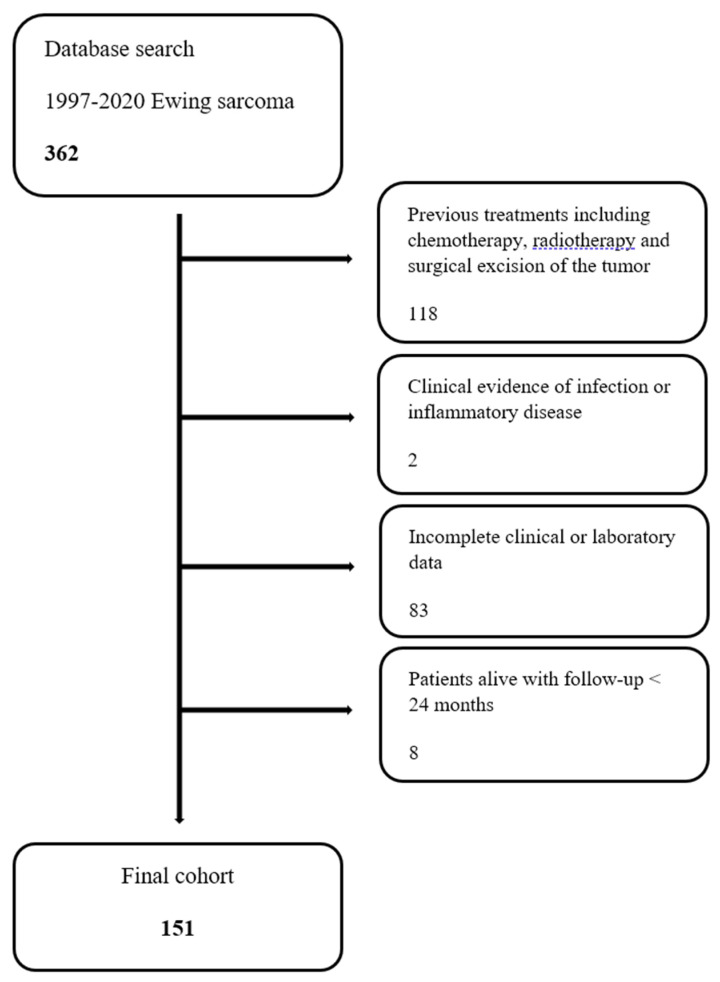
Flow diagram of children with Ewing’s sarcoma treated at our institution during the time of study.

**Figure 2 cancers-15-01573-f002:**
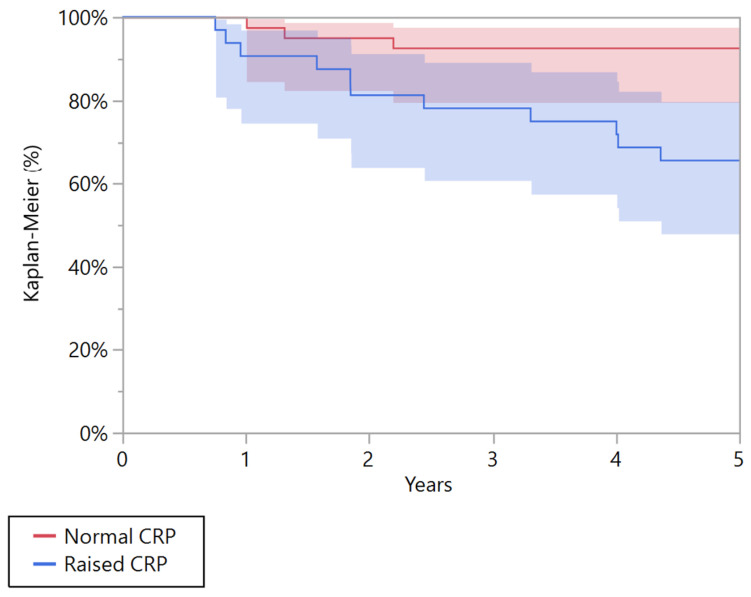
Kaplan-Meier curve showing the overall survival for patients affected by localized Ewing’s sarcoma with normal and raised C-reactive protein (CRP).

**Figure 3 cancers-15-01573-f003:**
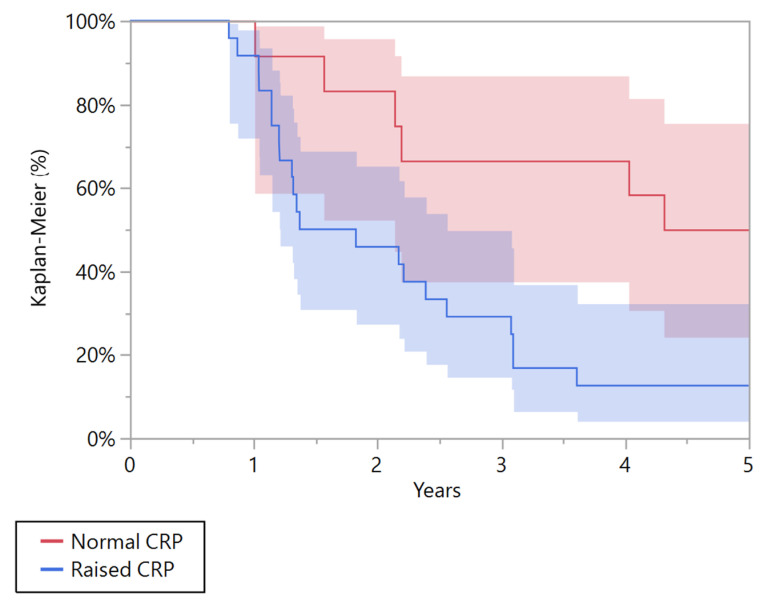
Kaplan-Meier curve showing the overall survival for patients affected by metastatic Ewing’s sarcoma with normal and raised C-reactive protein (CRP).

**Figure 4 cancers-15-01573-f004:**
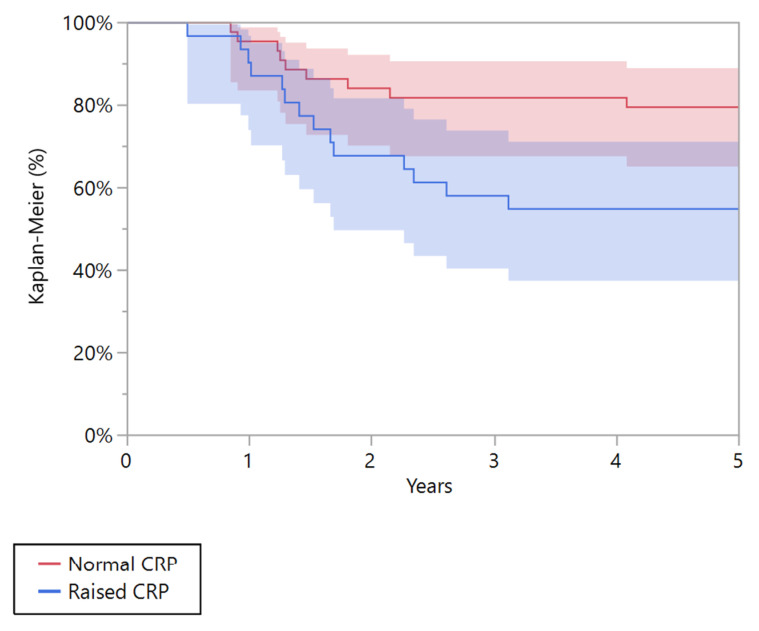
Kaplan-Meier curve showing the recurrence-free survival for patients affected by localized Ewing’s sarcoma with normal and raised C-reactive protein (CRP).

**Table 1 cancers-15-01573-t001:** Patient data.

Variable	Localized	Metastatic	Overall
Patients (*n*)	108	43	151
Gender (*n*)			
Male	79 (73.1%)	32 (74.4%)	111 (73.5%)
Female	29 (26.9%)	11 (25.6%)	40 (26.5%)
Age (years)	12.7 (3–18)	13.9 (6–18)	13.1 (3–18)
Tumor location (*n*)			
Extremities	81 (75.0%)	26 (60.5%)	107 (70.9%)
Trunk	27 (25.0%)	17 (39.5%)	44 (29.1%)
Tumor size (*n*)			
>200 mL	31 (28.7%)	29 (67.4%)	60 (39.7%)
<200 mL	70 (64.8%)	9 (20.9%)	79 (52.3)
missing	7 (6.5%)	5 (11.7%)	12 (7.9%)
Surgery (*n*)			
Yes	95 (88.0%)	19 (44.2%)	114 (75.5%)
No	13 (22.0%)	24 (55.8%)	37 (24.5%)
Radiotherapy (*n*)			
None	83 (76.9%)	15 (34.9%)	98 (64.9%)
Exclusive	13 (12.0%)	21 (48.8%)	34 (22.5%)
Before surgery	6 (5.5%)	2 (4.6%)	8 (5.3%)
Adjuvant	6 (5.5%)	5 (11.6%)	11 (7.3%)
Neoajuvant CHT (*n*)			
Yes	108 (100%)	42 (97.7%)	150 (99.3%)
No	0	1 (2.3%)	1 (0.7%)

**Table 2 cancers-15-01573-t002:** Prognostic factors associated with overall survival in children with localized or metastatic disease.

	Localized			Metastatic			Overall		
Variables	Raised CRP	Normal CRP	*p*-Value	Raised CRP	Normal CRP	*p*-Value	Raised CRP	Normal CRP	*p*-Value
*n*. Patients (%)	37	60	*p* = *0.025* *(Proportion Z-Test)*	28	12	*p = 0.017* *(Proportion Z-Test)*	65 (43.0)	72 (47.7)	*p = 0.608* *(Proportion Z-Test)*
Gender *n*. (%)			*p = 0.238* *(Fisher’s Exact Test)*			*p = 1.000* *(Fisher’s Exact Test)*			*p = 0.249* *(Fisher’s Exact Test)*
Male	30 (81.1)	41 (68.3)		21 (75.0)	9 (75.0)		51 (78.5)	50 (69.4)	
Female	7 (18.9)	19 (31.7)		7 (25.0)	3 (25.0)		14 (21.5)	22 (30.6)	
Mean age (min-max)	13.4 (3.0–18.0)	12.4 (3.0–18.0)	*p = 0.224* *(t-Test)*	14.5 (6.0–18.0)	12.3 (6.0–18.0)	*p = 0.055* *(t-Test)*	13.8 (3.0–18.0)	12.3 (3.0–18.0)	*p = 0.021* *(t-Test)*
*n*. Tumor location (%)			*p = 0.020* *(Fisher’s Exact Test)*			*p = 0.152* *(Fisher’s Exact Test)*			*p < 0.001* *(Fisher’s Exact Test)*
Extremities	24 (64.9)	52 (86.7)		15 (53.6)	10 (83.3)		39 (60.0)	62 (86.1)	
Trunk	13 (35.1)	8 (13.3)		13 (46.4)	2 (16.7)		26 (40.0)	10 (13.9)	
Tumor size			*p = 0.004* *(Fisher’s Exact Test)*			*p = 0.027* *(Fisher’s Exact Test)*			*p < 0.001* *(Fisher’s Exact Test)*
>200 mL	16 (43.2)	10 (16.7)		22 (78.6)	5 (41.7)		38 (58.5)	15 (20.8)	
<200 mL	18 (48.6)	46 (76.7)		3 (10.7)	5 (41.7)		21 (32.3)	51 (70.8)	

**Table 3 cancers-15-01573-t003:** Univariate Kaplan-Meier analysis of C-reactive protein (CRP) as a prognostic factor for overall survival and recurrence-free survival.

	Localized			Metastatic		
	Pathological CRP	Normal CRP	Log Rank Test	Raised CRP	Pathological CRP	Log Rank Test
5 years overall survival (95% CI)	65.6% (47.9–79.8)	92.7% (79.6–97.6)	0.004	12.5% (4.1–32.4)	50.0% (24.4–75.6)	0.010
5 years recurrence-free survival (95% CI)	54.8% (37.4–71.1)	79.5% (65.1–89.0)	0.024	13.0% (4.3–33.5)	45.5% (20.3–73.2)	0.076

**Table 4 cancers-15-01573-t004:** Univariate Kaplan-Meier analyses of laboratory biomarkers and clinical parameters for overall survival at 5 years.

Variables	*n*	Death in 5 Years	*p*-Value (Log-Rank Test)
Sex			0.771
Male	70	16	
Female	19	5	
Tumor location (*n*)			0.595
Extremities	75	17	
Trunk	14	4	
Tumor size			0.309
>200 mL	30	8	
<200 mL	49	9	
CRP			0.003
Pathological	34	14	
Normal	47	6	
Monocyte count			0.482
Raised	50	13	
Normal	39	8	
Neutrophile count			0.445
Raised	8	3	
Normal	80	18	
Lymphocyte count			0.199
Raised	16	3	
Normal	73	18	
Haemoglobin			0.570
Low	22	6	
Normal	67	15	
LDH			0.264
Raised	22	7	
Normal	54	14	
ALP			0.993
Raised	9	2	
Normal	80	19	
NLR			0.054
Raised	34	12	
Normal	55	9	
PLR			0.634
Raised	55	14	
Normal	34	7	
Radiotherapy			0.010
Yes	15	7	
No	74	14	
Metastasis at diagnosis			<0.0001
Yes	19	11	
No	70	10	

**Table 5 cancers-15-01573-t005:** Multivariate Cox regression model of metastasis at diagnosis, radiotherapy and C-reactive protein (CRP) for overall survival at 5 years.

Variables	HR (95% CI)	*p*-Value (Log-Rank Test)
Metastasis at diagnosis	4.27 (1.58–11.47)	0.004
Radiotherapy	1.7 (0.57–4.86)	0.310
Pathological CRP	3.67 (1.46–10.42)	0.005

**Table 6 cancers-15-01573-t006:** Univariate Kaplan-Meier analyses of laboratory biomarkers and clinical parameters associated with disease recurrence.

Variables	*n*	Recurrence in 5 Years	*p*-Value (Log-Rank Test)
Sex			0.999
Male	74	27	
Female	20	7	
Tumor location (*n*)			0.184
Extremities	79	27	
Trunk	15	7	
Tumor size			0.736
>200 mL	31	11	
<200 mL	41	17	
CRP			0.002
Pathological	35	19	
Normal	49	11	
Monocyte count			0.531
Raised	52	20	
Normal	42	14	
Neutrophile count			0.305
Raised	10	5	
Normal	84	29	
Lymphocyte count			0.467
Raised	6	3	
Normal	88	31	
Haemoglobin			0.955
Low	23	8	
Normal	71	26	
LDH			0.576
Raised	23	9	
Normal	66	23	
ALP			0.995
Raised	9	3	
Normal	85	31	
NLR			0.026
Raised	36	18	
Normal	58	16	
PLR			0.803
Raised	57	21	
Normal	37	13	
Radiotherapy			0.050
Yes	17	9	
No	77	25	
Metastasis at diagnosis			0.001
Yes	19	12	
No	75	22	

**Table 7 cancers-15-01573-t007:** A multivariate Cox regression model of metastasis at diagnosis, radiotherapy and C-reactive protein (CRP) associated with disease recurrence.

Variables	HR (95% CI)	*p*-Value (Log-Rank Test)
Metastasis ad diagnosis	2.56 (1.13–5.55)	0.024
Radiotherapy	1.86 (0.69–4.49)	0.201
Pathological CRP	2.66 (1.23–6.01)	0.012
NRL	1.72 (0.79–3.74)	0.165

## Data Availability

The data analyzed can be found in the electronic medical records of the patients who participated in the study.

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
