# Peer review of "The Prognostic Value of the Serum Level of C-Reactive Protein for Survival of Children with Ewing’s Sarcoma"

_cancers, 2023, doi:10.3390/cancers15051573_

Round 1

Reviewer 1 Report

The authors in this manuscript have tried to know the prognostic value of the serum level of C-reactive protein for survival of children with Ewing's sarcoma.

Better planned study in large cohort will be great to do and the comments below will make this manuscript better and comprehensive.

1. The manuscript needs improvement in English and scientific language.

2. Methods part needs to be more explanatory and how CRP was measured.

3. Adding Kaplan Meier plot survival curves will be great and the tables needs to be arranged to have a better view and understanding for the readers.

Author Response

Dear Reviewer,

1) We made an English editing.

2) The AU Beckman Coulter 680 analyzer was used for the CRP analysis and Dasit system XN-1000 for the remaining laboratory biomarkers [7].

3) We added Kaplan Meier plot survival curves.

Thanks to your suggestions we believe our article is better.

Reviewer 2 Report

This is a well-designed study which has been diligently performed. The manuscript  is concisely written and the data are clearly presented, with a higher patient number (n=151) mostly confirming previous findings. My major concern refers to the missing discussion of some previous publications on the  topic, which originate from different countries including Italy, with two of them being published in Cancers: 

Consalvo S, Hinterwimmer F, Harrasser N, Lenze U, Matziolis G, von Eisenhart-Rothe R, Knebel C. C-Reactive Protein Pretreatment-Level Evaluation for Ewing's Sarcoma Prognosis Assessment-A 15-Year Retrospective Single-Centre Study. Cancers (Basel). 2022 Nov 29;14(23):5898. doi: 10.3390/cancers14235898. PMID: 36497377; PMCID: PMC9735882.

Del Baldo G, Abbas R, De Ioris MA, Di Ruscio V, Alessi I, Miele E, Mastronuzzi A, Milano GM. The Prognostic Role of the C-Reactive Protein and Serum Lactate Dehydrogenase in a Pediatric Series of Bone Ewing Sarcoma. Cancers (Basel). 2022 Jun 22;14(13):3064. doi: 10.3390/cancers14133064. PMID: 35804835; PMCID: PMC9264769.

Aggerholm-Pedersen N, Maretty-Kongstad K, Keller J, Baerentzen S, Safwat A. The Prognostic Value of Serum Biomarkers in Localized Bone Sarcoma. Transl Oncol. 2016 Aug;9(4):322-8. doi: 10.1016/j.tranon.2016.05.006. PMID: 27567955; PMCID: PMC5006814.

These publications should be quoted and appropriately discussed. In particular conflicting findings should be addressed such as the finding by Consalvo et al. based on n=82 that elevated CRP is associated with systemic but not with local relapse (Cancers 2022).

Author Response

Dear Reviewer,

We added the two articles you suggested, analyzing the results and limitations. The authors did not analyze tumor size, other inflammatory biomarkers (only CRP) and did not consider subgroup analysis of localized patients and metastatic patients. In addition, they used 0.5 mg/dL instead of 1 mg/dL as a cut off for physiological CRP, unlike recent literature regarding prognostic inflammatory biomarkers for cancers and sarcomas.

Thanks to your suggestions we believe our article is better.

Round 2

Reviewer 2 Report

The authors now discussed the missing references appropriately.